# Association of Intradialytic Hypotension and Ultrafiltration with AKI-D Outcomes in the Outpatient Dialysis Setting

**DOI:** 10.3390/jcm11113147

**Published:** 2022-06-01

**Authors:** Emaad M. Abdel-Rahman, Ernst Casimir, Genevieve R. Lyons, Jennie Z. Ma, Jitendra K. Gautam

**Affiliations:** 1Division of Nephrology, University of Virginia, Charlottesville, VA 22908, USA; ecasimir1993@gmail.com (E.C.); jkg8h@hscmail.mcc.virginia.edu (J.K.G.); 2Department of Public Health Sciences, University of Virginia, Charlottesville, VA 22908, USA; grl2b@virginia.edu (G.R.L.); jzm4h@virginia.edu (J.Z.M.)

**Keywords:** acute kidney injury, end stage kidney disease, dialysis, mortality

## Abstract

Identifying modifiable predictors of outcomes for cases of acute kidney injury requiring hemodialysis (AKI-D) will allow better care of patients with AKI-D. All patients with AKI-D discharged to University of Virginia (UVA) outpatient HD units between 1 January 2017 to 31 December 2019 (n = 273) were followed- for up to six months. Dialysis-related parameters were measured during the first 4 weeks of outpatient HD to test the hypothesis that modifiable factors during dialysis are associated with AKI-D outcomes of recovery, End Stage Kidney Disease (ESKD), or death. Patients were 42% female, 67% Caucasian, with mean age 62.8 ± 15.4 years. Median number of dialysis sessions was 11 (6–15), lasting 3.6 ± 0.6 h. At 90 days after starting outpatient HD, 45% recovered, 45% were declared ESKD and 9.9% died, with no significant changes noted between three and six months. Patients who recovered, died or were declared ESKD experienced an average of 9, 10 and 16 intradialytic hypotensive (IDH) episodes, respectively. More frequent IDH episodes were associated with increased risk of ESKD (*p* = 0.01). A one liter increment in net ultrafiltration was associated with 54% increased ratio of ESKD (*p* = 0.048). Optimizing dialysis prescription to decrease frequency of IDH episodes and minimize UF, and close monitoring of outpatient dialysis for patients with AKI-D, are crucial and may improve outcomes for these patients.

## 1. Introduction

Acute kidney injury (AKI), defined by KDIGO in 2012 by serum creatinine and urinary output criteria [1], is a clinical syndrome characterized by rapid (within hours to days) impairment of kidney function. It is associated with serious consequences, both short and long term. AKI extends beyond affecting the kidneys to potentially affect other organs, thus causing further complications such as: increased cardiac events (congestive heart failure, coronary artery disease and stroke), recurrent hospitalizations, and catheter-related infections [2,3].

AKI requiring dialysis (AKI-D) is the most severe form of AKI, with prevalence reported to be 1–2% in hospitalized patients and 5–13% in critically ill patients [4,5,6]. Its incidence is on the rise, with increasing by 10% over the decade between 2000 and 2009 [7]. Globally, 41.2% of hospitalized AKI patients remain dialysis-dependent upon their hospital discharge [8].

Patients with AKI-D discharged from hospital to an outpatient hemodialysis setting may recover to become dialysis-independent, or may continue to have long-term renal complications which require prolonged dialysis [9,10,11], end stage kidney diseases (ESKD), or even death [5,6,12].

We [10,13] and others have reported improved outcomes when AKI-D patients were dialyzed in specialized AKI units, as required by the Centers for Medicare and Medicare Services (CMS) 2012 policy. Beginning on 1 January 2017, CMS reversed their 2012 policy, thus allowing outpatient ESKD facilities to return to furnishing dialysis services to AKI-D patients.

This change in policy, in addition to the high global burden of AKI-D and the need for evidence-based clinical guidance, makes it vital to identify predictors of AKI-D outcomes in order to guide further policy and clinical care. A 2015 editorial emphasized the lack of evidence-based clinical guidance and the need for increased information extraction from clinical and research databases, to positively affect aspects of health care such as patient mortality, quality of life, and cost containment [14]. A more recent review article re-emphasized the need for continuous research and multidisciplinary approaches aiming to improve AKI patient outcomes [15].

Predicting outcomes for these patients with AKI-D will serve patients, clinicians, and health communities at large, as it may allow clinicians to target therapeutic needs and offer timely alternative long-term therapeutic options to patients at critical junctures. To illustrate, if a patient is predicted to remain dialysis-dependent, then subsequent care could be better planned, patient education about kidney replacement therapy (KRT) options could be improved, referral for vascular access versus peritoneal catheter could be better assessed, and renal transplantation evaluations would be initiated after earlier referrals. On the other hand, if a patient is more likely to recover and become dialysis-independent, better measures could be taken to ensure adequate care of residual kidney function, to avoid hypotensive episodes on dialysis, to avoid nephrotoxic drugs, to more optimally monitor residual renal function, to establish closer nephrology outpatient follow-up once off dialysis, and to more efficiently coordinate care between different stakeholders. 

While multiple demographic and comorbid variables were identified as predictors of AKI-D outcome, many of these parameters are non-modifiable [9,10,13,16,17]. Modifiable factors that may play a role in predicting and hence promoting better care of AKI-D include the dialysis prescription and the dialysis procedure in the outpatient setting. However, most studies assessing the role of dialysis prescriptions on AKI-D patient outcomes have been conducted in the inpatient setting [18,19,20,21,22].

In the absence of clear guidelines for dialyzing patients with AKI-D in the outpatient setting, the overarching goal of these studies is to help guide policies to enable ESKD units to perform more standardized optimal dialysis for these patients. In this present study, we aimed to test the hypothesis that modifiable factors during dialysis sessions are associated with 90-day outcomes of recovery, ESKD, and death. Specifically, we hypothesized that poorer outcomes (ESKD and death) were associated with frequent drops in blood pressure during dialysis sessions, mean arterial pressure, and ultrafiltration rate.

## 2. Materials and Methods

### 2.1. Study Design and Population

This retrospective study examines the outcomes of AKI-D patients dialyzed at one of the eleven University of Virginia (UVA) outpatient dialysis units for three and six months post-discharge from hospital between 1 January 2017–31 December 2019. 

### 2.2. Methods

All AKI-D patients were dialyzed using our previously published protocol {Gautam, 2015, Predictors and Outcomes of Post-Hospitalization Dialysis Dependent Acute Kidney Injury}. The following data were obtained: non-modifiable clinical and demographic factors including age, gender, race, prior AKI episodes, existence of comorbidities including diabetes mellitus (DM), congestive heart failure (CHF), coronary heart disease (CAD), hypertension (HTN), as well as baseline kidney function, defined using last known serum creatinine measured before diagnosis with AKI, assessed by estimated glomerular filtration rate (eGFR). 

Dialysis data obtained included dialysis blood pressures (BP) before, during and after dialysis, ultrafiltration rate, mean arterial pressure (MAP), pulse, and weight. These data were used to calculate net ultrafiltration and number of hypotensive episodes per session for each patient. Data from the first four weeks of dialysis were analyzed. 

The primary covariate of interest was number of intradialytic hypotensive (IDH) episodes. Using the patient BP recordings obtained about every 30 min during a 3–4 h long dialysis session, we defined an intradialytic hypotensive episode as per KDIGO guideline [23]. Number of IDH episodes was summed across sessions in the first four weeks for each patient. The patients were further classified into quartiles according to number of IDH episodes, and the outcome performances among these four groups were evaluated in regression models. In addition, MAP was also considered.

Net ultrafiltration (UF) volume and rate at each session were calculated in liters and ml/kg/hour respectively. Net UF volume was defined as change in weight in kg (post-pre dialysis), while the UF rate at each session was defined as percentage weight change divided by session length in hours. The UF rate, Net UF, and MAP change were calculated for each session for each patient, and averaged across that patient’s dialysis sessions in the first four weeks of dialysis. These averages were used as predictors in the regression models.

Outcomes studied were recovery, declaration of ESKD, or death. Recovery of kidney function following AKI-D is defined by the Acute Disease Quality Initiative (ADQI) as sustained independence (>14 days) from kidney replacement therapy (KRT) [24]. 

### 2.3. Statistical Analysis

Dialysis recovery (off dialysis), ESKD, or death during the follow-up period were the primary AKI outcomes of interest. Association of patient characteristics at baseline and dialysis parameters from the first 4 weeks was evaluated in multinomial logistic regression with the nominal AKI outcomes. Both 3- and 6-month follow-up time frames were considered. At 3 months post start hemodialysis in the outpatient setting, 28 patients died, 122 patients recovered enough kidney function to be off dialysis and 123 patients continued to require dialysis. None of the patients who had renal recovery at 3 months required further hemodialysis and none of the patients who were dialysis-requiring at 3 months had enough renal recovery to be taken off dialysis. Thus, results at 6 months remained the same.

Univariate analyses were performed first followed by the final multivariable model that included those covariates with *p*-value < 0.1 from the univariate analysis. Because Net UF and UF rate are highly correlated and contain similar information, only Net UF was included in the final model.

A LOESS plot was used to graphically examine the relationship between number of hypotensive episodes and logit probability of ESKD vs. recovery (excluding those who died). This plot suggested a linear relationship between hypotensive episodes and probability of ESKD, thus, a separate analysis was conducted using the binary endpoint ESKD vs. recovery (n = 245). This allowed estimation of the increase in odds of ESKD for each additional hypotensive episode. Unadjusted and adjusted odds ratio (ORs) were estimated.

Complete case analysis was used. The final adjusted model included 206 patients after excluding incomplete cases. Baseline eGFR was missing for 45 patients and prior AKI was missing for 28 patients. Hypertension status was missing for 14 patients.

### 2.4. Ethical Considerations

This work was approved by the University of Virginia (UVA) Institutional Research Ethics Committee, IRB # 22068.

## 3. Results

AKI-D patients (n = 273) were 42% female, 67% Caucasian, with mean age 62.8 ± 15.4 years. Comorbidities included DM (42%), HTN (78%), CHF (18%), CAD (27%), and prior AKI episodes (36%), with pre AKI eGFR 33.8 ± 29.1 mL/min. Over a maximum of 28 days, the median (IQR) number of dialysis sessions was 11 (6–15), lasting 3.6 ± 0.6 h. All patients had dialysis access by tunneled central catheters, using anticoagulants with adequate blood and dialysate flow rates. Characteristics of all patients studied and characteristics of patients based on their outcomes are shown in Table 1. 

At 90 days post start of outpatient HD, 45% recovered, 45% were declared ESKD and 9.9% died. Between three and six months post start of outpatient HD, two more patients recovered, two patients died, and one patient who was initially off HD was declared ESKD. Patients declared ESKD had a higher number of median IDH episodes (16) in the first 4 weeks of dialysis than those who recovered (9) or died (10). 

In the final adjusted model, number of IDH episodes, net UF, and UF rate were associated with ESKD (Table 2). Patients with more frequent IDH episodes (i.e., those in the third and fourth quartiles) had increased odds of ESKD compared to patients in the lowest quartile. Adjusted odds ratios (95% CIs) for ESKD were 3.8 (1.4–9.8, *p* < 0.01) and 2.7 (1.0–7.9, *p* = 0.05) for patients in third and fourth quartiles, respectively (Figure 1). The odds ratio for Net ultrafiltration (UF) (Liters) was 1.5, so for each additional liter of UF we observed about a 54% increase in odds of ESKD (95% CI: 1.0–2.4, *p* = 0.0484) (Figure 2). None of the other dialysis variables were associated with our outcomes. Prior AKI was associated with about twice the odds of ESKD (aOR 2.1, 95% CI: 1.1–4.1).

In an unadjusted logistic model for ESKD versus recovery (n = 245), the odds ratio for continuous number of BP drops was 1.03 (95% CI: 1.01–1.05), corresponding to an estimated 3% increase in ESKD risk for each additional drop in blood pressure (*p* = 0.0008). After adjusting for age, previous AKI, HTN, and pre-AKI kidney function, the odds ratio remained marginally significant at 1.02 (95% CI: 1.0–1.5), *p* = 0.049. 

In the final multinomial logistic model, we adjusted for blood pressure drops, prior AKI, age, pre-AKI EGFR, HTN, and net UF. Since net UF and UF rate are well correlated, as expected, and showed similar association, we chose to include only net UF. In this model, only age was associated with death, with about a 6% increase in odds of death for each additional year of age. 

In an analysis using 2 weeks of dialysis data instead of 4 weeks, we did not observe statistically significant results, although the same trends were observed (data not shown). 

Using a chi-square test and a nonparametric Wilcoxon Mann-Whitney test to examine whether the quartile of number of blood pressure drops and the number of blood pressure drops, respectively, were associated with CHF. No association was detected, with *p*-values of 0.5811 and 0.6365, respectively.

## 4. Discussion

While several publications have highlighted the roles of demographic and comorbid variables, laboratory parameters, medications, and biomarkers, in predicting AKI-D patient outcomes. However, the roles of modifiable factors, such as dialysis prescription and procedure, in predicting and hence improving outcomes of these patients have rarely been studied. Furthermore, most of the previous studies looking at dialysis impact on outcomes were carried out in the inpatient setting rather than in the outpatient setting. Several parameters that may affect IDH and AKI-D outcomes in the inpatient setting have been studied, including the dialysis modality, type of dialysate and the dialyzer membrane used. RRT can be continuous, intermittent, or hybrid, as in cases of prolonged intermittent RRT (PIRRT). Studies examining the effect of dialysis modality on IDH have been controversial, with some studies showing superior hemodynamic parameters for CRRT over intermittent HD [25,26], and other studies showing no difference [27,28,29].

Our study has shown that increased frequency of IDH episodes is associated with increased incidence of ESKD. Specifically, the adjusted model suggested an increase in the odds of ESKD for patients in the third and fourth quartiles of number of blood pressure drops. In our data, this corresponded to 12 or more blood pressure drops over 4 weeks of dialysis. To inform clinical practice, we initially looked for a cutoff or maximum safe number of episodes, but because the relationship between number of drops and logit probability of ESKD was approximately linear, no obvious cutoff point emerged from the data. Because we observed significantly higher odds of ESKD in patients in the third and fourth quartiles of number of episodes of hypotension, we can hypothesize that the threshold is perhaps in this range. 

In agreement with our results, Pajewski et al. [30], analyzed data from 100 consecutive patients with AKI who survived to hospital discharge and required outpatient dialysis. Data was obtained in the first week post hospital discharge and showed that net fluid removal (5.3 vs. 4.1 L, *p* = 0.037), higher ultrafiltration rates (6.0 vs. 4.7 mL/kg/h, *p* = 0.041) and more frequent intradialytic hypotension (24.6% vs. 9.3% with 3 or more episodes, *p* = 0.049) correlated with absence of renal recovery. Our study went beyond one week post hospital discharge, analyzing four weeks of data. On analyzing two weeks of data, though we observed a trend for frequency of IDH episodes and net UF to correlate with outcomes, statistical significance was not achieved, raising the question of extent of follow-up time needed before making a clinical decision. 

Follow up beyond the first three months did not show significant changes in patient outcomes in our sample, though it may be reasonable to continue to monitor patients with AKI-D for signs of recovery for up to six months. 

Intradialytic hypotension in patients with AKI-D has been reported to occur in 30–87.3% of cases [31,32,33]. Several mechanisms have been suggested to explain the intradialytic hypotensive episodes and hemodynamic instabilities associated with hemodialysis in these patients. In a review article, Douvris et al. [22] identified several factors related to dialysis that may contribute to IDH in these patients: excessive UF, rapid osmotic shifts, dialysate flow rate, myocardial stunning, temperature changes during renal replacement therapy (RRT), dialysate fluid composition, dialyzer bio-incompatibility, mode of clearance, and vasopressor clearance by RRT. Many of the aforementioned mechanisms are subject to clinical intervention and can thus be modified to improve individual patients’ outcomes; given a standard of optimization.

Schortgen et al. [34] compared two cohorts of AKI-D patients dialyzed at a medical intensive care unit (ICU) during different time periods (1995 vs. 1997). They noted significantly fewer hypotensive episodes and less need for therapeutic interventions in the 1997 cohort (n = 76) vs. the 1995 cohort (n = 45), with no changes in mortality. They outlined several approaches that may have contributed to the improvement of hemodynamic instability during RRT. Their approach included the use of modified cellulosic membranes instead of unmodified cellulose, dialysate sodium concentration set to 145 mmol/L or higher, maximal blood flow rate of 150 mL/min, minimum session duration of 4 h, and dialysate temperature of 37 °C or lower. They further suggested a change in dialysis prescription to include initiating the HD session with dialysis and continuing with UF alone, or initiating the session without UF, and then adapting the UF rate to the hemodynamic response in more unstable patients. 

Edrees et al. [35] confirmed the value of cooler dialysate in decreasing episodes of hypotension in patients with AKI-D (n = 21) treated in the intensive care unit (ICU) and a step-down unit. On the other hand, Du Cheyron et al. [21] studied ICU patients with AKI-D (n = 74) randomized to a standard protocol using cooled and high sodium dialysate versus using blood-volume-controlled or blood-volume- and blood-temperature-controlled dialysis, but failed to show any significant impact of controlled body temperature and UF profiled by online monitoring of the incidence of intradialytic hypotensive episodes in these patients. These conflicting publications feed into the complexities of the various causes of intradialytic hypotensive episodes, and how the many variables involved may not necessarily garner the same response from different individuals when specifically modified.

Douvris et al. [36] identified five RCT and four observational studies that assessed the impact of any intervention effects on hemodynamic instability related to RRT in AKI-D patients managed in a medical or surgical ICU (n = 623 patients). Interventions included dialysate sodium modeling, UF profiling, blood and temperature control, duration of RRT, and slow blood flow rate. Out of these interventions, only higher dialysate sodium concentrations, lower dialysate temperature and variable UF rates were shown to reduce hemodynamic instability in the patients studied.

In our present study, we have further shown that higher net ultrafiltration rate during dialysis sessions was associated with increased incidence of ESKD. This finding is not surprising, yet it is beneficial to note when dialyzing patients with AKI-D in an outpatient dialysis setting. It is also helpful to highlight that for each additional liter of UF we observed about a 54% increase in the odds of ESKD. This data can help guide policy for AKI-D dialysis prescription in outpatient settings, complemented by the fact that lower net UF correlates with a lower incidence of intradialytic hypotensive episodes and thus improved patient outcomes on multiple levels.

Murugan et al. undertook two studies on the same cohort of critically ill AKI-D patients treated with CRRT in Australia and New Zealand (n = 1433) to determine the association between net UF rate and time to renal recovery [18,19]. The studies showed that higher net UF (>1.75 mL/kg/hour) correlated with more time to renal recovery and lower incidence of patient survival. On the other hand, Wu et al. [20] showed that positive fluid balance rather than ultrafiltration volume or ultrafiltration rate was more prognostic of 30 days mortality in hospitalized AKI-D patients after cardiac surgery (n = 63). 

Our study had several limitations; its retrospective design, omission of urine output analysis when calculating the correlation of net UF with outcomes, and of the roles played by medications in influencing the outcomes. Its strength includes studying AKI-D in the outpatient setting to suggest some recommendations to help in the formation of a more standardized protocol for dialysis prescription and monitoring patients’ hemodynamic status. Data collection over four weeks is another strength of the study. Though all units followed the same protocol, we cannot rule out some deviations from the protocol in one or more of the units studied. 

Our study has the potential to influence clinical practice. It suggests that development of predictive models to predict recovery could be a fruitful direction. Earlier prediction of outcomes can suggest changes to clinical practice that will enhance recovery for certain groups of patients.

## 5. Conclusions

Recovery of renal function among patients with AKI-D in the outpatient setting is a reachable goal, with significant numbers of patients becoming dialysis-independent within 90 days post hospital discharge. A standardized protocol that allows for limiting ultrafiltration goals and the number of intradialytic hypotensive episodes may promote renal recovery of patients with AKI-D dialyzed in the outpatient setting.

## Figures and Tables

**Figure 1 jcm-11-03147-f001:**
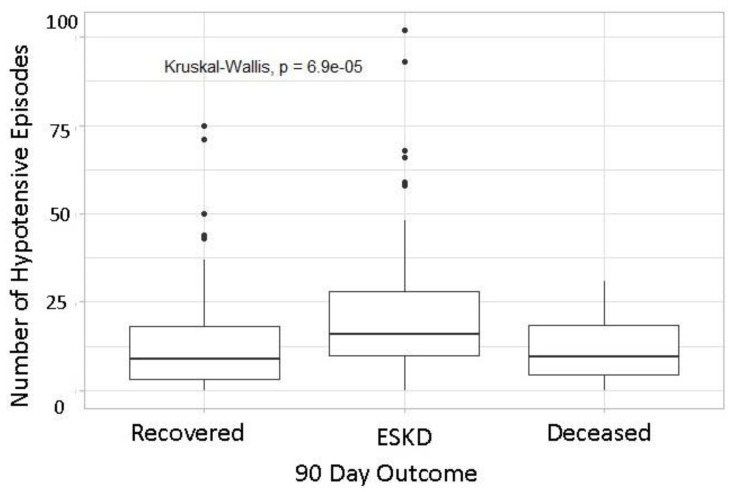
Analysis of number of Intra Dialytic Hypotensive (IDH) episodes across all sessions in the first 4 weeks. ESKD: End Stage Kidney disease.

**Figure 2 jcm-11-03147-f002:**
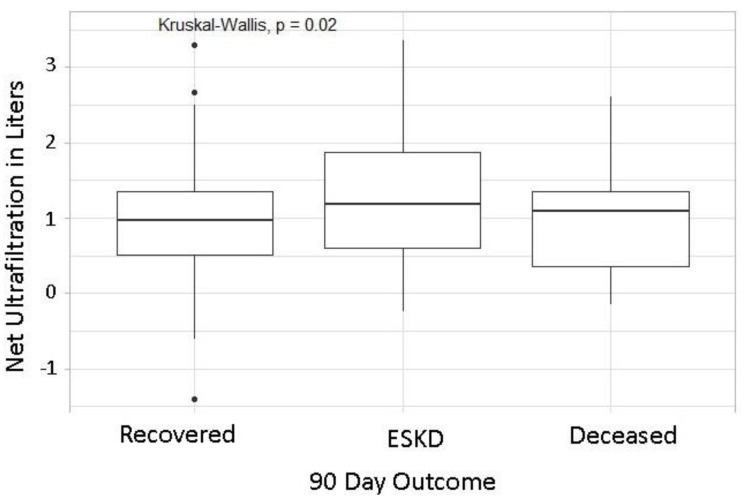
Analysis of Net Ultrafiltration across all sessions. ESKD: End Stage Kidney disease.

**Table 1 jcm-11-03147-t001:** Study Patient Characteristics.

Characteristic		Total N = 273	Deceased N = 28 (%)	ESKD ^&^ N = 123 (%)	Recovered N = 122 (%)
Sex	Female	115	13 (46.4)	54 (43.9)	48 (39.3)
	Male	158	15 (53.6)	69 (56.1)	74 (60.7)
Race	American Indian	1	0 (0)	1 (0.8)	0 (0)
	Black	72	7 (25.0)	34 (27.6)	31 (25.4)
	Hispanic	6	0 (0)	6 (4.9)	0 (0)
	Other	8	2 (7.1)	3 (2.4)	3 (2.5)
	While	182	19 (67.9)	78 (63.4)	85 (69.7)
	Missing	4	0 (0)	1 (0.8)	3 (2.5)
CHF *	No	211	22 (78.6)	98 (79.7)	91 (74.6)
	Yes	49	5 (17.9)	23 (18.7)	21 (17.2)
	Missing	13	1 (3.6)	2 (1.6)	10 (8.2)
CAD ^#^	No	185	20 (71.4)	82 (66.7)	83 (68.0)
	Yes	75	7 (25)	39 (31.7)	29 (23.8)
	Missing	13	1 (3.6)	2 (1.6)	10 (8.2)
Hypertension	No	47	2 (7.1)	18 (14.6)	27 (22.1)
	Yes	213	25 (89.3)	103 (83.7)	85 (69.7)
	Missing	13	1 (3.6)	2 (1.6)	10 (8.2)
Prior AKI	No	147	15 (53.6)	59 (48.0)	73 (59.8)
	Yes	99	10 (35.7)	56 (45.5)	33 (27.0)
	Missing	27	3 (10.7)	8 (6.5)	16 (13.1)
Diabetes	No	158	16 (57.1)	66 (53.7)	76 (62.3)
	Yes	115	12 (42.9)	57 (46.3)	46 (37.7)

^&^ End Stage Kidney disease (ESKD); * Chronic Heart Failure (CHF); ^#^ Coronary Artery Disease (CAD).

**Table 2 jcm-11-03147-t002:** Multinomial Logistic Regression Model Results.

	Odds Ratio Estimates						
**Covariate**	**90 Day Outcome**	**Odds Ratio**	**95% Confidence Interval**	**Covariate**	**90 Day Outcome**	**Odds Ratio**	**95% Confidence Interval**
Hypotensive Episodes, 2nd quartile vs. 1st	**Deceased**	1.1	0.3–4.1	Hypotensive Episodes, 2nd quartile vs. 1st	**ESKD ***	2.3	1.0–5.7
Hypotensive Episodes, 3rd quartile vs. 1st	**Deceased**	1.2	0.3–4.6	**Hypotensive** **Episodes, 3rd** **quartile vs. 1st**	**ESKD**	**3.8**	**1.4–9.8**
Hypotensive Episodes, 4th quartile vs. 1st	**Deceased**	0.5	0.1–2.5	**Hypotensive** **Episodes, 4th** **quartile vs. 1st**	**ESKD**	**2.7**	**1.0–7.9**
Prior AKI, Yes vs. No	**Deceased**	1.8	0.6–4.8	**Prior AKI**, **Yes** **vs. No**	**ESKD**	**2.1**	**1.1–4.1**
**Age**	**Deceased**	**1.1**	**1.0–1.1**	Age	**ESKD**	1.0	1.0–1.0
Baseline Kidney Function	**Deceased**	1.0	1.0–1.0	Baseline Kidney Function	**ESKD**	1.0	1.0–1.0
Hypertension, Yes, vs. No	**Deceased**	2.5	0.5–12.2	Hypertension, Yes, vs. No	**ESKD**	1.4	0.9–3.2
Net Ultrafiltration, Liters	**Deceased**	1.0	0.5–2.0	**Net** **Ultrafiltration**, **Liters**	**ESKD**	**1.5**	**1.0–2.4**

* End Stage Kidney disease (ESKD).

## Data Availability

Data supporting reported results can be found in the electronic archived datasets at the University of Virginia.

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
