# Peer review of "Association of Intradialytic Hypotension and Ultrafiltration with AKI-D Outcomes in the Outpatient Dialysis Setting"

_jcm, 2022, doi:10.3390/jcm11113147_

Round 1

Reviewer 1 Report

Thank you for writing the manuscript with the title: »Outpatient dialysis prescription as predictor and modifiable factor for outcomes of patients with AKI-D«. The authors present a compelling article about AKI and its outcomes, underlying the detrimental effect of intradialytic hypotension episodes on the AKI outcome. The article is very well written and informative and it also carries important weight as it can influence the clinical practice in various settings.

I would like to suggest the authors to add some data and expand on the potential role of dialysis modality (CRRT vs PIRRT vs IHD) in intradialytic hypotensive episodes and their potential effect on the outcome. 

Author Response

Response to critique: Reviewer 1:

I would like to suggest to the authors to add some data and expand on the potential role of dialysis modality (CRRT vs PIRRT vs IHD) in intradialytic hypotensive episodes and their potential effect on the outcome

 **Thank you so much for this critique. Though the inpatient dialysis is beyond the scope of this manuscript as the study addresses the outpatient dialysis setting where all patients were on intermittent hemodialysis (IHD), we added a paragraph (Lines 210-216) concerning the potential role of dialysis modality on IDH.

Reviewer 2 Report

Outpatient dialysis prescription as a predictor and modifiable 2 factors for outcomes of patients with AKI-D

Thank you for selecting an interesting and useful topic.

The title does not represent the content of the manuscript. Most of the parameters like IDH are multifactorial.

Results can be better presented. 1st table needs to revise (Eg 1st cell male: female proportions are wrong).

There are limitations to having controlled samples in a retrospective study. Nevertheless, giving some information such as the definition of AKI, how HD prescription is decided, and the definition of “recovery” help to generalize your findings.

Mean pre-AKI eGFR of 33.8 (+/- 29.1) indicate majority if not all, of the sample, had AKI on CKD. Hence, could these results be applicable to non-CKD-AKI is doubtful.

Were you able to identify any association between IDH and CHF? If so please mention it. Indeed, it will help in the prevention of high-risk groups.

Author Response

Response to Reviewer 2

Thank you so much for your thorough and thoughtful review

  • The title does not represent the content of the manuscript

Response:

** Thank you so much. Title changed

  • Table 1 needs to be revised

Response:

**Agree. Table 1 revised and corrected.

  • Giving some information such as definition of AKI, how hD prescription is decided, and the definition of recovery

Response:

**Definition of AKI added (lines 30-31), definition of AKI recovery (lines 119-120) is already in the text. HD prescription was individualized per dialysis unit.

  • Mean pre-AKI eGFR of 33.9 (+/- 29.1) indicate majority if not all, of the sample, had AKI on CKD. Hence, applicability of these results in non –CKD- AKI is doubtful

 Response:

** Agree.

  • Were you able to identify any association between IDH and CHF?

 Response:

** Association between IDH and CHF was added (lines 1980201). No significant association

Reviewer 3 Report

This article describes that Acute Kidney Injury (AKI) patients requiring dialysis are at increased risk of death or developing End Stage Kidney Disease (ESKD) if they have hypotensive events or high volumes of net ultrafiltration.
The manuscript has numerous critical issues for which rejection is suggested:

1) The abstract is written with many acronyms which make it incomprehensible on first reading.

2) The methodological section is extremely lacking: with what definition are these patients considered affected by AKI? Furthermore, patients were not stratified by disease severity and causes of disease such as sepsis are not known. How many patients were on antibiotics or vasoactive drugs? How many patients developed multi-organ failure among the three groups? 3)

3)there is no novelty. The development of shock and the need for renal replacement therapy (RRT) are two severity characteristics of AKI and are associated with poor prognosis, so it can be deduced that groups of patients with hypotensive events and the need for RRT will have higher mortality and ESKD in the future.

Author Response

  • The abstract is written with many acronyms which make it incomprehensible on first reading.

Response:

**Acronyms explanation added

  • The methodological section is extremely lacking: with what definition are these patients considered affected by AKI? Furthermore, patients were not stratified by disease severity and causes of disease such as sepsis are not known. How many patients were on antibiotics or vasoactive drugs? How many patients developed multi-organ failure among the three groups?

Response:

** Agree with reviewer. These are limitations of a retrospective trial. But our main objective was to test whether parameters as intradialytic hypotension and ultrafiltration may affect outcomes of patients discharged to an outpatient facility.

Definition of AKI is mentioned in the text based on KDIGO definition 2012.

  • There is no novelty. The development of shock and the need for renal replacement therapy (RRT) are two severity characteristics of AKI and are associated with poor prognosis, so it can be deduced that groups of patients with hypotensive events and the need for RRT will have higher mortality and ESKD in the future

Response:

** The objective of the manuscript is to impress among outpatient nephrologists that managing AKI is different than managing ESRD patients and that limiting IDH episodes and UF rates may positively impact patients’ outcomes.

Round 2

Reviewer 3 Report

I thank the authors for editing according to the suggestions. However, I reject the manuscript because, as admitted by the authors themselves, it has many limitation. The main confounder is the stage of disease which the authors for which the authors did not stratify the population. Patients who have higher IDH and UF are those who are likely to have a more severe stage of disease and therefore risk of ESKD and death.

Author Response

Thank you so much for this critique. All patients are AKI-D patients, the most severe form of AKI, so stratifying patients based on the stage of the AKI severity cannot be done.

I do agree with the reviewer that stratifying the patients based on co-morbid may have been helpful. On reviewing the literature, Pajewski et al (Hemodialysis International 2018; 22:66-73) showed that several comorbidities was associated with AKI outcomes (Recovery vs non-recovery) in the inpatient setting. Increase age, higher Charlson comorbidity score and lower baseline eGFR were associated with non-renal recovery after AKI. On the other hand, outcome in the outpatient setting correlated with albumin levels, UF and IDH. Furthermore, in our study (Gautam et al. Nephron 2015;131:185-190) looking at patients with AKI-D , none of the following comorbidities (DM, CHF, ICU admission, peripheral vascular diseases,  hypertension, presence of proteinuria) correlated with outcomes in the outpatient setting. Similarly, Rathore et al (Nephron 2017;137:105-112) showed the same findings with the above comorbidities as well as no correlation was detected between sepsis and the use of vasopressors and outcome of patients with AKI-D in the outpatient setting.

Lastly, the manuscript was focused on modifiable factors that can affect outcome of AKI-D patients in the outpatient setting, the presence of comorbidities, etiology of AKI and the use of vasopressors while inpatient are not modifiable.
